# Inflammatory Myofibroblastic Tumor: An Updated Review

**DOI:** 10.3390/cancers17081327

**Published:** 2025-04-15

**Authors:** Joon Hyuk Choi

**Affiliations:** Department of Pathology, Yeungnam University College of Medicine, Daegu 42415, Republic of Korea; joonhyukchoi@ynu.ac.kr

**Keywords:** inflammatory myofibroblastic tumor, inflammatory pseudotumor, epithelioid inflammatory myofibroblastic sarcoma, molecular genetics, tumor classification

## Abstract

Inflammatory myofibroblastic tumor (IMT) is a rare neoplasm that is characterized by the proliferation of myofibroblastic and fibroblastic spindle cells interspersed with a mixed inflammatory infiltrate that is predominantly composed of plasma cells and lymphocytes. Recent advancements have improved the understanding of molecular biology and therapeutic strategies for IMT. This review comprehensively analyzed the clinical, pathological, and molecular genetic characteristics of IMT, highlighting diagnostic approaches and crucial differential considerations.

## 1. Introduction

Inflammatory myofibroblastic tumor (IMT) is a distinctive neoplasm with limited metastatic potential, characterized by the proliferation of fibroblastic and myofibroblastic spindle cells, accompanied by a dense inflammatory infiltrate that is primarily composed of plasma cells, lymphocytes, and eosinophils [1]. IMT predominantly occurs in children and young adults, particularly in the abdominal cavity, retroperitoneum, and lungs. The prognosis is generally favorable but is affected by the anatomical location of tumors and the feasibility of complete surgical resection [2].

Currently, IMT is the generally accepted term encompassing lesions that are previously classified under various reactive and neoplastic entities, including plasma cell granuloma [3,4], plasma cell pseudotumor [5], inflammatory myofibrohistiocytic proliferation [6], omental mesenteric myxoid hamartoma [7], inflammatory pseudotumor (IPT) [8,9,10], inflammatory fibrosarcoma [11,12], and inflammatory myofibroblastic sarcoma [13,14]. Significant morphological and clinical overlaps caused this classification unification, supported by accumulating clinical and genetic evidence—particularly anaplastic lymphoma kinase (*ALK*) rearrangements—confirming the neoplastic nature of these lesions [15]. Consequently, the 2020 World Health Organization (WHO) classification of soft tissue tumors (fifth edition) no longer recommends using these alternative terms.

Precise pathological classification and a comprehensive understanding of IMT’s molecular pathogenesis are crucial for improving patient treatment and prognosis. Recent advancements in molecular genetic techniques have significantly enhanced our understanding of IMT biology, classification, prognosis, and treatment strategies [16]. However, IMT diagnosis and treatment remain challenging due to its significant histological and molecular heterogeneity. This review provides a comprehensive update on IMT pathology, highlighting molecular genetics, histological features, and key differential diagnoses.

## 2. Epidemiology

IMT is a rare mesenchymal neoplasm, accounting for less than 1% of all soft tissue tumors. Its exact prevalence is difficult to determine due to its rarity, histological heterogeneity, and historical misclassification as a reactive or inflammatory process. IMT predominantly affects children and young adults but occurs across a wide age spectrum [17,18]. There is a slight female predominance; however, tumor location may impact both the age of onset and gender distribution. IMT most commonly arises in the lung, abdominal cavity—particularly the mesentery and omentum—and the retroperitoneum. However, IMT can also occur in various other locations, including the gastrointestinal tract, pelvis, mediastinum, genitourinary tract, and head and neck [19,20].

Gastrointestinal IMTs predominantly involve the small intestine and colon, followed by the stomach. Less commonly, they affect the esophagus, appendix, pancreas, and liver. The submucosa, muscularis propria, and mesentery may also be affected [21].

Thoracic IMTs can arise in the lung, tracheobronchial tree, pleura, chest wall, and mediastinum. Pulmonary IMTs account for up to 1% of all lung tumors [22]. They constitute approximately 50% of all benign or intermediate pulmonary neoplasms in children [23,24]. The male-to-female ratio is approximately 1:1.

Although IMT can arise in various sites within the genitourinary system, the urinary bladder is the most commonly affected organ [25,26,27]. Urinary bladder IMTs primarily occur in middle-aged adults but can also develop in children, accounting for approximately 25% of reported cases [28,29]. ALK-positive tumors are slightly more prevalent in females and tend to occur at a younger age than ALK-negative tumors [26].

IMTs of the female genital tract are rare, with approximately 100 cases reported [30]. They predominantly occur in adult, premenopausal women, with a median age of 38 years (range, 6–78 years) [31,32]. Some cases have been associated with pregnancy and may involve the placenta [33,34,35,36,37]. In the female genital tract, IMTs most commonly arise in the uterine corpus [38,39], with less frequent involvement in the cervix [31,32].

Head and neck IMTs predominantly affect adults but can occur across a wide age range, from infancy to 81 years [40,41,42,43,44]. No consistent sex predilection has been observed [45,46]. Approximately 15–25% of IMTs arise in the head and neck, most commonly in the larynx (true vocal cords), sinonasal tract (maxillary sinus), and oral cavity [47,48]. Other affected sites include the trachea, pharynx, orbit, skull base, salivary glands, and neck.

## 3. Clinical and Radiological Features

### 3.1. Clinical Features

The clinical presentation of IMTs varies depending on the site of origin [49,50]. Abdominal IMTs may present with vague abdominal pain, gastrointestinal obstruction, or bleeding. Peritoneal tumors often manifest as an abdominopelvic mass, whereas pulmonary IMTs are commonly associated with cough, chest pain, and dyspnea. Uterine IMTs typically present with abnormal uterine bleeding, whereas urinary bladder IMTs frequently cause hematuria, dysuria, or abdominal pain. Laryngeal IMTs often lead to hoarseness and dysphonia, whereas sinonasal and skull base IMTs may result in nasal obstruction, pain, visual disturbances, and cranial nerve palsies.

Approximately 20–30% of patients with IMTs present with systemic inflammatory syndrome characterized by fever, malaise, weight loss, and abnormal laboratory findings. These laboratory abnormalities include microcytic hypochromic anemia, thrombocytosis, polyclonal hypergammaglobulinemia, and elevated erythrocyte sedimentation rate and C-reactive protein levels [6,51]. This syndrome typically resolves after tumor resection but may recur if the tumor relapses.

### 3.2. Radiological Features

IMTs exhibit highly variable imaging characteristics, reflecting differences in their fibrous and cellular composition [52]. IMTs typically present as lobulated, heterogeneous solid masses with or without calcification (Figure 1). However, radiological features vary depending on the anatomic site. Pulmonary IMTs typically present as peripheral, well-defined lobulated masses, predominantly in the lower lobes [53]. These lesions demonstrate heterogeneous enhancement on computed tomography (CT). In contrast, head and neck IMTs exhibit non-specific imaging findings; however, an indistinct, invasive mass with heterogeneous enhancement on CT and MRI may indicate IMT [54]. Retroperitoneal IMTs typically reveal an isointense to slightly hypointense signal on T1-weighted images and a hyperintense signal on T2-weighted images, depending on the cellular composition, and are usually accompanied by heterogeneous contrast enhancement. Differentiating IMT from inflammatory lesions and other soft tissue neoplasms based solely on imaging findings is challenging. Accurate diagnosis requires integration of imaging with clinical evaluation and laboratory test results.

## 4. Histopathological Features

### 4.1. Macroscopic Features

IMT typically presents as a well-circumscribed, multinodular mass. Most cases are solitary, whereas multiple nodules are observed in approximately one-third of cases [55]. Tumor size varies considerably, ranging from 1 cm to over 20 cm, with a median size of 5–6 cm. The tumor appears white to gray, yellow, or tan on gross examination and may present a whorled, fleshy, or myxoid texture (Figure 2).

### 4.2. Histopathology

Histologically, IMTs exhibit diverse morphological features characterized by spindle-shaped and elongated fibroblastic and myofibroblastic cell proliferations. These tumors demonstrate significant variation in cellularity, stromal composition, and inflammatory infiltrate. The tumor cells typically possess vesicular nuclei, small nucleoli, and pale eosinophilic cytoplasm. Nuclear atypia is generally minimal; however, cases with severe atypia may be observed. IMTs are classified into three distinct histological patterns: (1) myxoid, (2) hypercellular, and (3) hypercellular fibrous [13,17,18]. The myxoid pattern is characterized by loosely arranged plump or spindled myofibroblasts in a myxoid background (Figure 3a). This pattern features abundant blood vessels and a prominent inflammatory infiltrate of plasma cells, lymphocytes, and eosinophils, resembling granulation tissue or a reactive process. The hypercellular pattern is defined by a dense, compact proliferation of fascicular spindle cells in a variable myxoid and collagenous stroma, accompanied by an inflammatory infiltrate (Figure 3b). The hypocellular fibrous pattern is distinguished by a hyalinized, collagen-rich stroma with reduced spindle cell density and a relatively sparse inflammatory infiltrate, resembling scar tissue or desmoid fibromatosis (DFM) (Figure 3c). Multiple histological patterns may coexist within a single tumor.

Notably, the inflammatory component is predominantly lymphoplasmacytic but may also be mixed, with variations in intensity and distribution within each tumor. Dystrophic calcifications and osseous metaplasia are occasionally observed. IMTs frequently exhibit a ganglion-like appearance, characterized by vesicular nuclei, prominent nucleoli, and abundant amphophilic or eosinophilic cytoplasm (Figure 3d). Additionally, epithelioid cells and Touton giant cells may also be present. Necrosis is uncommon. Mitotic activity varies but is generally low; however, mitoses may be numerous. Rare cases with overtly sarcomatous morphology have been reported, often associated with aggressive clinical behavior [27]. Table 1 summarizes the histological patterns and atypical features of IMT.

## 5. Immunohistochemical Features

Immunohistochemically, ALK immunoreactivity is detectable in approximately 50–60% of cases, exhibiting cytoplasmic, membranous, perinuclear, or dot-like staining patterns, which vary depending on the specific *ALK* fusion partner [56]. Many *ALK* fusion variants exhibit a diffuse cytoplasmic staining pattern, the most common ALK immunostaining pattern observed in IMTs (Figure 4). The use of highly sensitive ALK antibody clones (5A4, D5F3) enhances ALK protein detection in IMTs [57,58]. IMTs harboring *ROS1* rearrangement typically demonstrate cytoplasmic expression of ROS1 [58,59]. p53 immunostaining generally shows a wild-type pattern [60]. IMTs display variable staining for smooth muscle actin (SMA), muscle-specific actin (MSA), calponin, and desmin, consistent with a myofibroblastic immunophenotype. Focal cytokeratin immunoreactivity is observed in approximately 30% of cases.

## 6. Molecular Features

Clonal cytogenetic rearrangements involving the *ALK* gene at chromosome band 2p23 are detected in approximately 50–60% of IMTs, particularly in children and young adults. These rearrangements lead to the fusion of the 3′-kinase region of the *ALK* with various partner genes, including *TPM3*, *TPM4*, *CLTC*, *CARS*, *SEC31L1*, *PPFIBP1*, *ATIC*, *EML4*, *PRKAR1A*, *FN1*, *DCTN1*, *LMNA*, *TFG*, and *HNRNPA1*, with additional fusion partners continually being identified [61,62,63,64,65,66,67,68]. Epithelioid inflammatory myofibroblastic sarcomas (EIMSs) typically harbor a *RANBP2*::*ALK* fusion, whereas a small subset carriers an *RRBP1*::*ALK* fusion [14,69]. IMTs with *ALK* genomic rearrangements demonstrate the activation and overexpression of the ALK C-terminal kinase domain, a phenomenon restricted to the neoplastic myofibroblastic component [70,71]. These *ALK* rearrangements drive tumorigenesis through the constitutive activation and overexpression of the ALK receptor tyrosine kinase, which promotes oncogenic signaling.

*ROS1* and *NTRK3* gene rearrangements are detected at similar frequencies in approximately 5–10% of ALK-negative IMTs. The reported fusion partners include *TFG*::*ROS1*, *YWHAE*::*ROS1*, and *ETV6*::*NTRK3* [58,59,72,73]. *ROS1* rearrangements result in the constitutive activation of the ROS1 receptor tyrosine kinase, driving oncogenesis; however, these alterations occur independently of *ALK* signaling. Additionally, rare IMT cases harbor *RET* or *PDGFRB* gene rearrangements [68,73]. Table 2 summarizes the molecular genetic features of IMT and EIMS.

IMTs demonstrate molecular and genetic heterogeneity influenced by both age and anatomical location. *ALK* rearrangements are infrequent in IMTs diagnosed in older adults, and other gene rearrangements, including *ROS1*, *NTRK3*, *RET*, and *PDGFRB,* are rare in individuals over 40 years of age [74]. Approximately one-third of IMTs harbor *ROS1* or *NTRK3* gene fusions, which are more frequently found in infants and children [75]. In uterine IMTs, *ETV6*::*NTRK3* or a *RET* fusion has been reported [76], although rarely, whereas *ROS1* and *PDGFRB* fusions have not yet been identified at this site. In urinary tract IMTs, most cases harbor an *FN1*::*ALK* fusion [77]. Recently, RNA-based fusion analysis detected an ALK-negative urinary bladder IMT with a novel *FN1*::*RET* gene fusion [60]. Among IMTs in the head and neck, the most commonly detected rearrangement involves *ALK* fused with *TIMP3*, whereas *ROS1* fusions are rare [78,79].

## 7. Inflammatory Myofibroblastic Tumor Subtypes and Recently Described Myofibroblastic Sarcoma

### 7.1. Pregnancy–Associated Inflammatory Myofibroblastic Tumor

Pregnancy–associated inflammatory myofibroblastic tumor (PAIMT) is an increasingly recognized IMT subtype, with fewer than 50 cases reported to date [80,81,82,83,84,85,86,87,88,89,90]. It is typically detected incidentally in pregnant women, often in the context of prenatal complications such as diabetes mellitus, hypertension, preeclampsia, or abnormal implantation [87]. Notably, two reported cases have been associated with twin gestations [83,87]. The origin of PAIMTs—whether maternal or fetal—was initially unclear; however, short tandem repeat (STR) analysis confirmed a maternal origin [35]. These tumors are typically expelled at delivery or during the immediate postpartum period. If not expelled as a detached mass, PAIMTs may remain adherent to the placental disc or extraplacental fetal membranes [88]. Although a subset of PAIMTs would be classified as intermediate-risk based on the recently proposed risk classification model [82], all reported cases have exhibited benign behavior.

Grossly, PAIMTs are well-circumscribed and generally small, ranging from 0.5 to 9.0 cm in size, with a white to tan-pink cut surface. Occasionally, they exhibit a whorled, gelatinous, or myxoid appearance; however, hemorrhage is rare. Histologically, three distinct growth patterns have been identified: (1) hypocellular and myxoid, (2) decidualized and cellular with a moderate myxoid matrix, and (3) smooth muscle-like with minimal myxoid stroma [37]. Cytologic atypia ranges from absent to mild, and mitotic activity is rare (up to 3 per 10 high-power fields [HPFs]). Ganglion-like cells, tumor cells with bizarre nuclei, multinucleated cells, and tumor necrosis may occasionally be present [87]. The inflammatory infiltrate varies from minimal to diffuse and is typically enriched with lymphocytes and plasma cells. Immunohistochemically, most PAIMPs exhibit ALK positivity (cytoplasmic and perinuclear staining) and express progesterone receptor, and CD10, with variable expression of smooth muscle markers and estrogen receptor. Nearly all ALK-positive PAIMTs demonstrate *ALK* fusions by fluorescence in situ hybridization (FISH) or RNA sequencing. However, a small subset of ALK-negative tumors harbors fusions involving *RET*, *ROS1*, or *INSR* [86,89]. The most common gene partners include *TIMP3* and *THBS1*, which play a role in endometrial remodeling during pregnancy [86,87,89].

### 7.2. Epithelioid Inflammatory Myofibroblastic Sarcoma

EIMS is a distinctive and aggressive IMT subtype characterized by epithelioid cells with vesicular nuclei and large nucleoli in a prominent myxoid stroma [14]. EIMS predominantly affects males, with a median age of onset of 39 years (range, 7 months–63 years). These tumors primarily arise intra-abdominally, most commonly in the mesentery and omentum, although they can rarely occur in the lung, stomach, and brain [90,91,92]. Approximately 50% of cases are multifocal at diagnosis. Patients typically present with abdominal pain and ascites. Molecularly, EIMS is associated with *RANBP2*::*ALK* or *RRBP1*::*ALK* gene rearrangements [14,62,69]. The prognosis is generally poor, as EIMS follows a more aggressive course.

Grossly, EIMS tumors range from 8 to 26 cm in size, with a median size of 15 cm. Histologically, EIMS is characterized by plump epithelioid tumor cells with vesicular chromatin, prominent nucleoli, and amphophilic or eosinophilic cytoplasm (Figure 5a). A minor spindle cell component may also be present. The tumor predominantly exhibits abundant myxoid stroma and a prominent neutrophilic infiltrate [14,62], whereas plasma cells are often absent. The median mitotic rate is 4 per HPF (range, 1–18 per HPF), and focal necrosis is observed in approximately 50% of cases. Immunohistochemically, all EIMS tumor cells are positive for ALK. EIMS with *RANBP2*::*ALK* is associated with a nuclear membranous staining pattern (Figure 5b), whereas EIMS with *RRBP1*::*ALK* demonstrates a perinuclear accentuated cytoplasmic pattern. Additionally, most tumors express desmin and CD30, with variable SMA expression.

### 7.3. Myxoid Inflammatory Myofibroblastic Sarcoma

Recently, Papke et al. [93] described a distinctive myofibroblastic sarcoma subtype, provisionally termed “myxoid inflammatory myofibroblastic sarcoma (MIMS)”. MIMS shares certain features with IMT; however, its more aggressive behavior and distinct histological characteristics support its classification as a separate entity. MIMS occurs at a median age of 37 years (range, 7–79 years) and exhibits an approximately equal sex distribution. The primary tumor site is predominantly the peritoneum, although other reported locations include the paratesticular region, retroperitoneum, upper extremity, chest wall, esophagus, and uterus. Approximately 50% of peritoneal tumors are multifocal, whereas tumors in other locations are typically unifocal.

Histologically, MIMS is characterized by bland to mildly atypical neoplastic myofibroblasts in a myxoid stroma, with prominent inflammatory infiltrates observed in approximately 90% of cases (Figure 6a,b). Most tumors exhibit delicate, branching stromal vessels and demonstrate an infiltrative growth pattern into adjacent nonneoplastic tissue (Figure 6c). Immunohistochemically, the tumor cells show variable expression of SMA, desmin, and CD30 (Figure 6d). ALK expression is observed in approximately 5% of cases; however, no *ALK* rearrangement has been identified. Genomic sequencing of 11 tumors revealed tyrosine kinase fusions in approximately 60% of cases, including *PDGFRB* rearrangements, *PML*::*JAK1*, or *SEC31A*::*PDGFRA* [93]. Despite its bland histological appearance, MIMS carries a significant risk of disseminated disease, particularly when arising in the peritoneum. Targeted therapies may be considered for patients with disseminated diseases. Table 3 summarizes IMT subtypes and recently described myofibroblastic sarcoma.

## 8. Diagnostic Approach

A systematic diagnostic approach is essential for the accurate diagnosis of IMTs. The first and most crucial step is a thorough histological assessment of hematoxylin and eosin (H&E)-stained sections at low magnification. Pathologists should carefully assess tumor cell morphology, architectural growth patterns, and stromal characteristics. Additionally, clinical history and radiological findings provide valuable diagnostic clues, aiding in differentiating IMTs from their mimics. Immunohistochemistry (IHC) for ALK is a useful first-line diagnostic tool for IMT. If ALK staining is negative and a definitive diagnosis remains uncertain, molecular assays for *ALK* can be performed to confirm the diagnosis. FISH for *ALK* gene rearrangements is widely used (Figure 7); however, false-negative results may occur, particularly in cases with abundant inflammation that obscures tumor cells or intrachromosomal inversions (e.g., *EML4*::*ALK* fusion), which FISH can sometimes miss due to inadequate probe spacing [39]. RNA-based fusion assays have demonstrated higher sensitivity in detecting *ALK* rearrangements, particularly in cases involving cryptic, complex, or intrachromosomal rearrangements that FISH may fail to detect. In ALK-negative cases, FISH or RNA-based fusion assays are recommended to identify *ROS1, NTRK3*, or other gene fusions [58,72,73]. If these tests are negative, IHC for ROS1 or additional molecular tests for non-*ALK* gene fusions may be useful [72].

## 9. Differential Diagnosis

The differential diagnosis of IMT is broad and includes several morphologic mimics, such as IPT, IgG4-related sclerosing disease (IGSD), pseudosarcomatous myofiboblastic proliferation (PMP), inflammatory fibroid polyp (IFP), nodular fasciitis (NF), DFM, inflammatory well-differentiated liposarcoma (IWDLPS), low-grade myofibroblastic sarcoma (LGMS), myxoinflammatory fibroblastic sarcoma (MIFS), inflammatory leiomyosarcoma (ILPS), embryonal rhabdomyosarcoma (ERMS), gastrointestinal stromal tumor (GIST), and Epstein–Barr virus-positive inflammatory follicular dendritic cell sarcoma (EFDCS). Distinguishing IMTs from their morphological mimics is crucial for accurate clinical behavior prediction and identifying patients who may benefit from targeted therapy with ALK inhibitors. Clinical presentation, tumor site, histological features, and immunohistochemical profile are crucial in differentiating IMTs from their mimics. Table 4 summarizes the differential diagnosis of IMT.

### 9.1. Inflammatory Pseudotumor

IPTs are nonneoplastic inflammatory lesions that arise due to various pathological factors, including infection, immune reactions, and post-traumatic responses. Histologically, IPTs consist of a heterogeneous population of inflammatory cells, fibroblasts, and myofibroblasts with variable collagen deposition. The inflammatory infiltrate typically comprises lymphocytes, plasma cells, or histiocytes. Unlike IMTs, IPTs lack known genetic alterations. Immunohistochemically, myofibroblasts in IPTs express SMA, but ALK is typically negative.

### 9.2. IgG4-Related Sclerosing Disease

IGSD is a fibro-inflammatory disorder characterized by a dense lymphoplasmacytic infiltrate, abundant IgG4-positive plasma cells, and fibrosis. IGSDs may closely mimic the sclerosing pattern of IMT [94]. IGSDs commonly arise in the pancreas, orbit, salivary glands, lacrimal glands, and retroperitoneum. Histologically, IGSD is characterized by dense lymphoplasmacytic infiltration, storiform fibrosis, and obliterative phlebitis. Immunohistochemically, IGSD demonstrates prominent IgG4-expressing plasma cells, with an IgG4-positive/IgG-positive plasma cell ratio significantly higher than that of IMT. Unlike IMT, IGSD is consistently negative for ALK.

### 9.3. Pseudosarcomatous Myofibroblastic Proliferation

PMP is a reactive myofibroblastic proliferation that mimics sarcomatous features [95]. Approximately 70% of patients have a history of prior instrumentation or surgery. PMPs primarily affect the genitourinary tract, most commonly the urinary bladder, and may exhibit morphologic overlap with IMT. Molecularly, PMPs commonly harbor recurrent *FN1*::*ALK* fusions. Histologically, PMPs consist of myofibroblasts arranged in a “tissue culture-like” pattern, with palely eosinophilic cytoplasm and delicate cytoplasmic processes. Approximately 70% of cases exhibit an infiltrative growth pattern extending through the muscularis propria. Inflammatory infiltrates may be present in PMP but are less prominent than in IMT. Immunohistochemically, PMP is positive for ALK in approximately 70% of cases but negative for ROS1. Additionally, PMP expresses SMA and desmin, with variable cytokeratin expression.

### 9.4. Inflammatory Fibroid Polyp

IFP is a benign, often polypoid, hypocellular fibroblastic neoplasm that predominantly affects the stomach and ileum [96]. Histologically, IFPs consist of bland spindle or stellate cells in a loose edematous or myxoid stroma, accompanied by prominent eosinophils and scattered lymphocytes. A characteristic feature is the whorling arrangement of proliferating cells around blood vessels. Immunohistochemically, IFPs are positive for CD34, with occasional expression of SMA and desmin. Molecularly, the majority of IFPs harbor activating *PDGFRA* mutations. IHC for PDGFRA can aid in confirming the diagnosis [97]. IMT can arise in the gastrointestinal tract and may be mistaken for IFP. However, IFP is distinguished by its stellate cells, reactive blood vessels, and a prominent inflammatory infiltrate, particularly rich in eosinophils.

### 9.5. Nodular Fasciitis

NF is a self-limiting mesenchymal neoplasm that predominantly arises in the subcutaneous tissue of the extremities and measures <3 cm in size [98]. NF is relatively common and can occur at any age, although it is more frequently observed in young adults. Molecularly, NF is characterized by recurrent *USP6* rearrangements. Histologically, NF consists of plump, uniform fibroblasts and myofibroblasts arranged in a tissue culture-like growth pattern. The stroma is variably myxoid, featuring microcystic changes and extravasated erythrocytes. Immunohistochemically, the tumor cells are positive for SMA and MSA, with occasional desmin expression. NF is consistently negative for ALK.

### 9.6. Desmoid Fibromatosis

DFM is a locally aggressive but non-metastasizing deep-seated (myo)fibroblastic neoplasm characterized by infiltrative growth and a strong predisposition for local recurrence [99]. DFM commonly arises in the extremities, abdominal cavity, retroperitoneum, abdominal wall, and chest wall. Molecularly, DFM is driven by somatic *CTNNB1* mutations or inactivating germline *APC* mutations. Histologically, DFM consists of uniform fibroblastic proliferation arranged in long, sweeping fascicles with collagen deposition and minimal inflammation. Small-caliber vessels with perivascular edema are frequently observed. Immunohistochemically, DFM is positive for SMA and MSA and demonstrates nuclear β-catenin expression in most cases, whereas ALK expression is absent.

### 9.7. Inflammatory Well-Differentiated Liposarcoma

IWDLPS is a subtype of WDLPS that may exhibit a prominent chronic inflammatory component [100,101]. IWDLPS primarily arises in the retroperitoneum and is more common in older adults. *MDM2* amplification is consistently present. Histologically, IWDLPS is characterized by a dense chronic inflammatory cell infiltrate, predominantly composed of T lymphocytes. Scattered, highly atypical cells with enlarged, hyperchromatic nuclei are observed (Figure 8a). Adipocytic differentiation is often clearly identifiable in other regions of the tumor. Immunohistochemically, the tumor cells are positive for MDM2 and CDK4 and negative for ALK (Figure 8b). These features help distinguish IWDLPS from IMT.

### 9.8. Low-Grade Myofibroblastic Sarcoma

LGMS is a rarely metastasizing mesenchymal neoplasm that often exhibits fibromatosis-like features [102,103]. It has a wide anatomical distribution, most commonly involving the extremities and the head and neck region, particularly the tongue and oral cavity. Histologically, LGMS consists of spindle-shaped tumor cells arranged in cellular fascicles or a storiform growth pattern. The tumor cells display at least focal moderate nuclear atypia (Figure 9a) and demonstrate a diffusely infiltrative growth pattern, often extending between skeletal muscle fibers. Immunohistochemically, the tumor cells show variable expression of SMA and desmin (Figure 9b). Unlike IMT, LGMS lacks a prominent inflammatory infiltrate and myxoid stroma and is negative for ALK.

### 9.9. Myxoinflammatory Fibroblastic Sarcoma

MIFS is an infiltrative, locally aggressive fibroblastic neoplasm that predominantly arises in the distal extremities [104]. It commonly arises in the distal acral extremities, particularly fingers and hands. Molecularly, MIFS harbors a (1;10)(p22;q24) translocation, resulting in the *TGFBR3*::*MGEA5* gene fusion [105,106,107], which is also shared with hemosiderotic fibrolipomatous tumors. Additionally, *BRAF* rearrangements are present in a subset of MIFS [108]. Histologically, MIFS consists of atypical fibroblastic cells with macronucleoli set in a variably myxoid and hyalinized matrix, accompanied by a mixed inflammatory infiltrate (Figure 10). Immunohistochemically, the tumor cells exhibit variable expression of CD34, SMA, and CD68. Compared with IMT, MIFS exhibits more significant nuclear atypia and predominant acral locations.

### 9.10. Inflammatory Leiomyosarcoma

ILMS is a malignant neoplasm characterized by smooth muscle differentiation, a prominent inflammatory infiltrate, and near-haploidization [109]. It primarily arises in deep soft tissue, most commonly in the lower limb, followed by the trunk, retroperitoneum, and lung. Histologically, ILMS exhibits a fascicular proliferation of variably atypical eosinophilic spindle cells, accompanied by a prominent, usually diffuse inflammatory infiltrate (Figure 11a). The inflammatory component predominantly consists of small lymphocytes, often mixed with plasma cells and histiocytes. Immunohistochemically, ILMS is generally diffusely positive for SMA, variably positive for desmin, and negative for ALK (Figure 11b). Differentiation from IMTs may be challenging. Compared with IMT, the tumor cells in ILMS exhibit cigar-shaped nuclei with blunt ends and are arranged in a more regular fascicular growth pattern. Cloutier et al. [110] recently proposed an alternative designation, “inflammatory rhabdomyoblastic tumor”, for ILMS cases expressing rhabdomyoblastic markers, including myogenin, MyoD1, and PAX7.

### 9.11. Embryonal Rhabdomyosarcoma

ERMS is a malignant soft tissue tumor that exhibits morphological and immunophenotypic features of embryonic skeletal muscle [111]. RMS is the most common soft tissue sarcoma in children and adolescents. Approximately 50% of ERMS cases arise in the head and neck region, including the orbit, while the remaining cases primarily develop in the genitourinary system [112,113]. Histologically, ERMS is composed of primitive round and spindle-shaped cells. Scattered differentiated rhabdomyoblasts are present. Immunohistochemically, the tumor cells are positive for desmin and exhibit heterogeneous nuclear staining for myogenin and MYOD1. Unlike IMTs, ERMS contains rhabdomyoblasts and is negative for ALK.

### 9.12. Gastrointestinal Stromal Tumor

GIST is a mesenchymal neoplasm with variable biological behavior, characterized by differentiation toward the interstitial cells of Cajal [114]. Sporadic GISTs can develop at any age but most commonly occur in the sixth decade of life (median age, 60–65 years) and exhibit a slight male predominance [115]. Succinate dehydrogenase (SDH)-deficient GISTs primarily arise in the stomach, are more prevalent in females, and typically affect younger patients [116,117,118]. Molecularly, most GISTs harbor *KIT* or *PDGFRA* mutations. Histologically, GISTs exhibit a broad morphological spectrum, composed of relatively monomorphic spindle or epithelioid cells. Inflammation is generally minimal or absent. Immunohistochemically, GISTs are positive for CD34, CD117 (KIT), and DOG1. A minor subset of GITSs may express SMA and desmin. SDH-deficient GISTs exhibit loss of SDH subunit B (SDHB) protein expression, regardless of which *SDH* gene is mutated [119,120,121]. GISTs may closely resemble IMTs; however, they are distinguished by their positivity for CD117 (KIT) and DOG1 and their negativity for ALK.

### 9.13. Epstein–Barr Virus-Positive Inflammatory Follicular Dendritic Cell Sarcoma

EFDCS is an indolent malignant neoplasm characterized by the neoplastic proliferation of follicular dendritic cells (FDCs), a prominent lymphoplasmacytic infiltrate, and a consistent association with Epstein–Barr virus (EBV). EFDCS arises almost exclusively in the liver or spleen. Reported genomic alterations include copy-neutral loss of heterozygosity at 5q and gain of the X chromosome [122,123]. Histologically, EFDCS is characterized by a proliferation of atypical spindle to oval cells with indistinct cell borders, vesicular nuclei, and prominent nucleoli, accompanied by a dense lymphoplasmacytic infiltrate. Immunohistochemically, the tumor cells are positive for FDC markers, including CD21, CD23, CD35, CXL13, D2-40, and CNA.42. EFDCS is EBV-positive, with EBV-encoded small RNA (EBER) detectable by in situ hybridization (EBER-ISH). Both EFDCS and IMT exhibit prominent inflammatory infiltrates. IHC for FDC markers is crucial in distinguishing EFDCS from IMT.

## 10. Treatment and Prognosis

### 10.1. Treatment

Complete surgical resection remains the primary treatment for IMT, particularly in cases of localized disease. Achieving negative surgical margins is ideal, as this significantly reduces the risk of recurrence. In cases of recurrence, re-excision is generally recommended when feasible. The National Comprehensive Cancer Network (NCCN) guidelines recommend ALK inhibitors as first-line systemic therapy for advanced-stage, recurrent, metastatic, or inoperable IMTs with *ALK* fusions [124,125,126]. Tyrosine kinase inhibitors (TKIs), such as crizotinib, have demonstrated significant efficacy in ALK-positive IMTs [127]. For resistance or disease progression, alternative TKIs, including ceritinib, alectinib, and lorlatinib, may be considered [128,129,130]. ALK-negative IMTs may harbor alternative molecular alterations involving *ROS1*, *PDGFRβ*, or *NTRK* fusions. *ROS1*-rearranged IMTs may respond to ROS1 inhibitors, such as crizotinib, highlighting the therapeutic relevance of identifying *ROS1* fusions in ALK-negative cases. Tumors with *NTRK* fusions have shown favorable responses to specific TKIs, such as entrectinib or larotrectinib [44,58,131]. Additionally, anti-inflammatory agents have been investigated as a potential treatment strategy for IMTs with prominent inflammatory infiltrates, particularly when surgery or targeted therapy is not feasible [132].

### 10.2. Prognosis

IMT is generally categorized as a neoplasm with intermediate malignant potential and a low risk of metastasis. ALK-negative IMTs may have a higher metastatic potential; however, ALK immunoreactivity does not correlate with recurrence [17,50,70]. The prognosis is generally favorable in pediatric patients, even in unresectable and ALK-negative cases [133,134]. Distant metastases are rare (<5%) and most commonly involve the lungs, brain, liver, and bone. EIMS is an aggressive subtype characterized by rapid recurrence, disseminated intra-abdominal disease, variable liver metastases, and a high mortality rate [14]. Approximately 25% of extrapulmonary IMTs recur, with recurrence rates partly influenced by anatomical site and resectability [135].

### 10.3. Risk Stratification Model for Uterine Inflammatory Myofibroblastic Tumor

A risk stratification model for uterine IMT was recently proposed [136], utilizing a scoring system based on clinicopathologic factors to categorize patients by risk. The clinicopathologic risk stratification score assigns one point for each of the following criteria: (1) age >45 years, (2) tumor size ≥5 cm, (3) mitotic activity ≥4 mitotic figures per 10 HPFs, and (4) infiltrative tumor borders. Based on the total score, patients are classified into three risk categories. Patients with 0 points are classified as low-risk. Those with 1–2 points fall into the intermediate-risk group, for whom next-generation sequencing (NGS) is recommended if available. If *ALK* fusion alone is detected, the patient remains in the low-risk category. However, additional pathogenic molecular alterations result in reclassification to high-risk. Patients with ≥3 points are directly classified as high-risk. This integrated stratification system incorporates clinicopathologic features and molecular testing to improve risk assessment and guide potential treatment decisions.

## 11. Future Perspectives

The comprehensive study of IMT has been challenging due to the difficulty of assembling large case cohorts [137,138,139]. Histologically, IMTs often exhibit significant overlap with other spindle cell tumors, making accurate diagnosis particularly difficult, especially in ALK-negative cases. A comprehensive molecular workup is essential for identifying reliable biomarkers to ensure a precise diagnosis.

From a therapeutic perspective, ALK inhibitors, such as crizotinib, have demonstrated efficacy in treating ALK-positive IMTs. However, treatment options for ALK-negative or refractory cases remain limited, highlighting the need for alternative targeted therapies. Moreover, reliable prognostic indicators for conventional IMT have not yet been established. Further research is crucial to advancing our understanding of IMT biology, refining prognostic assessment, and optimizing treatment strategies, ultimately leading to more effective personalized therapies.

## 12. Conclusions

IMT is a rare myofibroblastic neoplasm that poses significant diagnostic and therapeutic challenges. Due to its considerable morphologic overlap with other tumors, accurate diagnosis requires immunohistochemical analysis and ancillary molecular testing. A thorough understanding of IMT, including its subtypes and clinical behavior, is crucial for guiding appropriate surgical management, optimizing the use of systemic therapy when indicated, and ensuring effective long-term follow-up. As our knowledge of IMT continues to evolve, the integration of molecular insights with advanced diagnostic and therapeutic strategies will be essential for improving patient outcomes and advancing precision medicine.

## Figures and Tables

**Figure 1 cancers-17-01327-f001:**
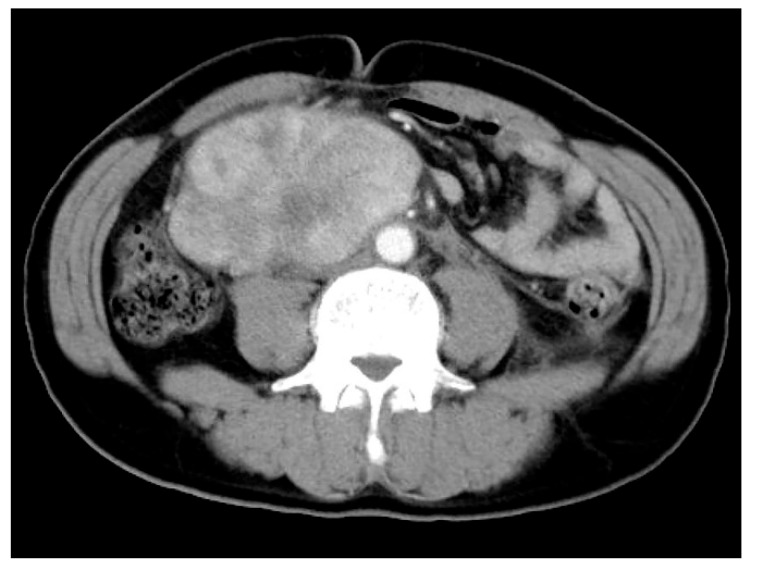
Radiological findings of inflammatory myofibroblastic tumor. Abdominal computed tomography shows a well-circumscribed, heterogeneously enhancing mass in the retroperitoneum.

**Figure 2 cancers-17-01327-f002:**
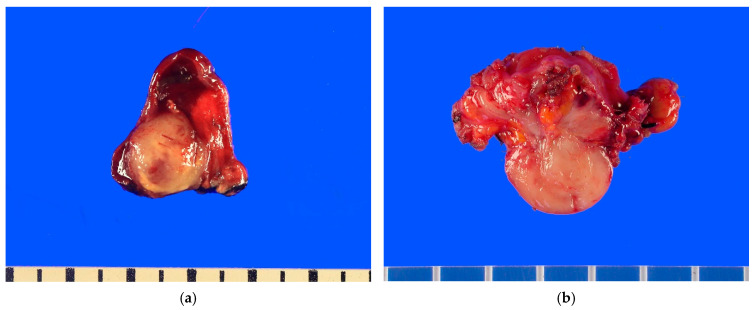
Macroscopic findings of inflammatory myofibroblastic tumor. (**a**) A well-circumscribed, yellow-gray to tan, solid mass is present in the lung. (**b**) A well-circumscribed, gray to white, solid mass is present in the wall of the cecum.

**Figure 3 cancers-17-01327-f003:**
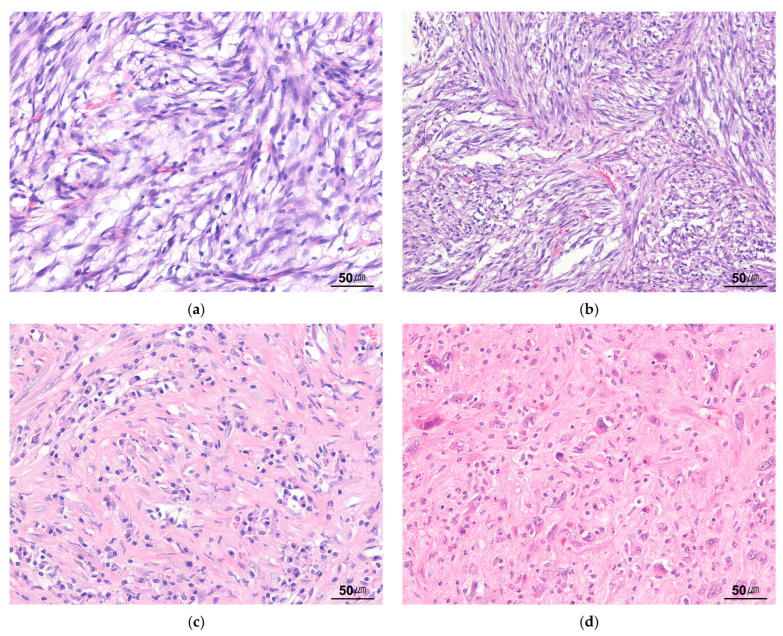
Histological findings of inflammatory myofibroblastic tumor. (**a**) Myxoid pattern shows spindled myofibroblasts dispersed in a myxoid stroma (H&E stain, ×200). (**b**) Hypercellular pattern shows a fascicular spindle cell proliferation (H&E stain, ×200). (**c**) Hypocellular fibrous pattern shows hyalinized collagenous stroma with sparse spindle cells and lymphoplasmacytic infiltrate (H&E stain, ×200). (**d**) Ganglion-like cells with abundant amphophilic cytoplasm, vesicular nuclei, and prominent nucleoli are present in the recurrent retroperitoneal inflammatory myofibroblastic tumor (H&E stain, ×200).

**Figure 4 cancers-17-01327-f004:**
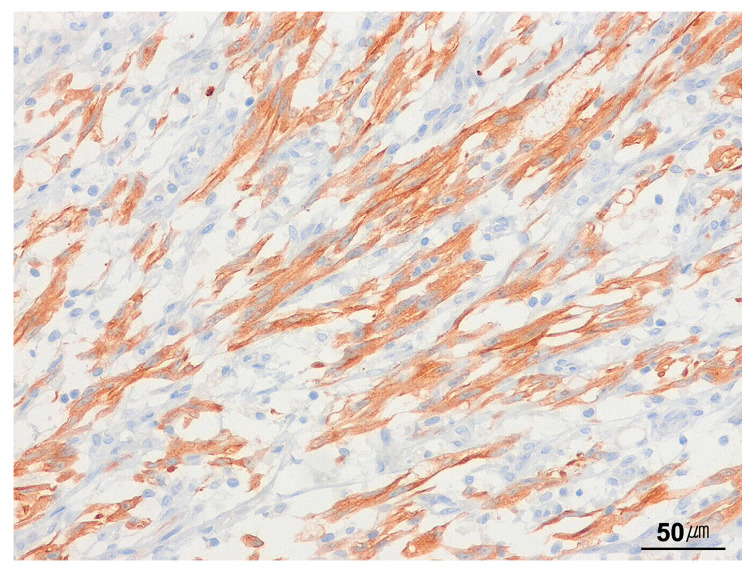
Immunohistochemistry for ALK in inflammatory myofibroblastic tumor. The tumor cells show diffuse cytoplasmic staining for ALK (immunohistochemical stain for ALK, ×200).

**Figure 5 cancers-17-01327-f005:**
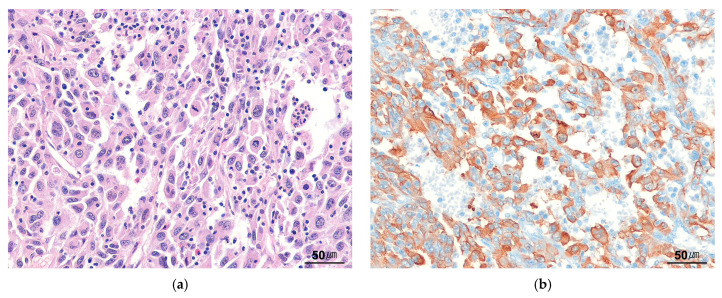
Epithelioid inflammatory myofibroblastic sarcoma. (**a**) Epithelioid tumor cells with abundant amphophilic cytoplasm are present. Neutrophilic infiltration is accompanied (H&E stain, ×200). (**b**) The tumor cells show perinuclear staining for ALK (immunohistochemical stain for ALK, ×200).

**Figure 6 cancers-17-01327-f006:**
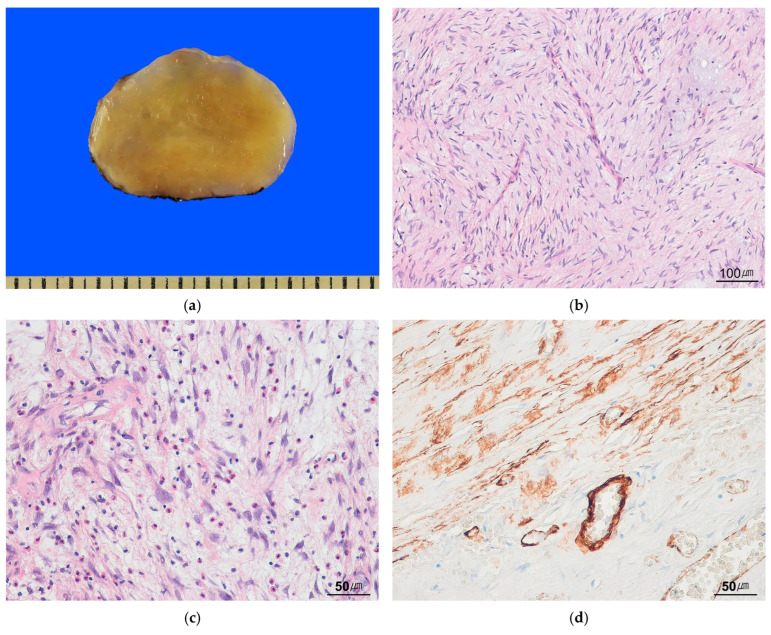
Myxoid inflammatory myofibroblastic sarcoma. (**a**) The tumor shows a well-circumscribed, yellowish myxoid cut surface. (**b**) Spindle-shaped tumor cells are present in a myxoid stroma with delicate blood vessels (H&E stain, ×100). (**c**) Mixed inflammatory cell infiltrates are present (H&E stain, ×200). (**d**) The tumor cells are positive for SMA (immunohistochemical stain for SMA, ×200).

**Figure 7 cancers-17-01327-f007:**
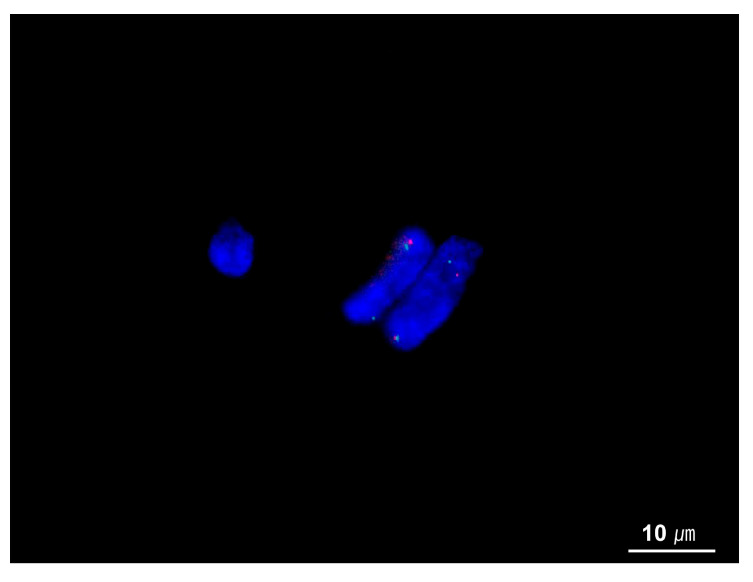
FISH for ALK of inflammatory myofibroblastic tumor. Break-apart FISH assays show *ALK* gene rearrangements with split green and red signals (FISH for ALK, ×1000).

**Figure 8 cancers-17-01327-f008:**
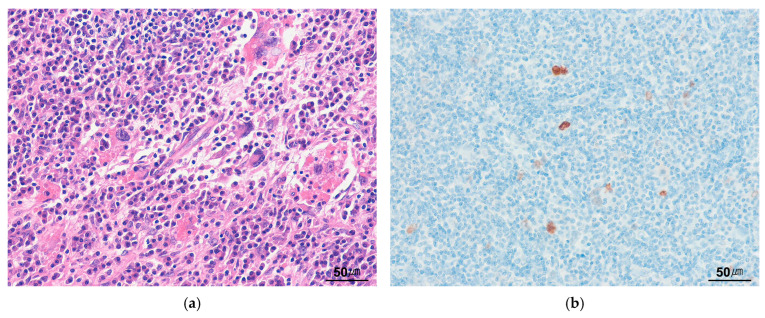
Inflammatory well-differentiated liposarcoma. (**a**) Atypical, multinucleated stromal cells are scattered, with prominent inflammatory cell infiltrate (H&E stain, ×200). (**b**) Atypical stromal cells are positive for MDM2 (immunohistochemical stain for MDM2, ×200).

**Figure 9 cancers-17-01327-f009:**
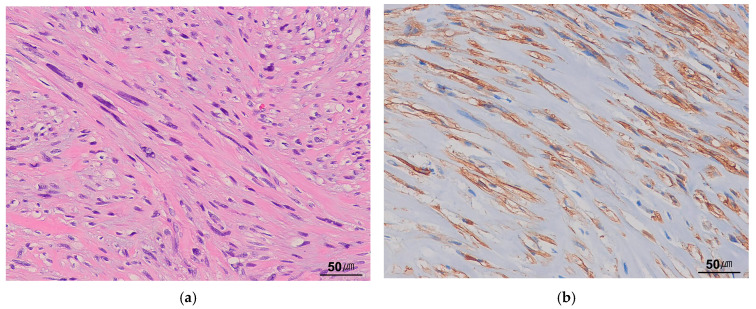
Low-grade myofibroblastic sarcoma. (**a**) The tumor is composed of cellular spindle cells with eosinophilic cytoplasm. Tumor cells with atypical hyperchromatic nuclei are often present (H&E stain, ×200). (**b**) The tumor cells are positive for desmin (immunohistochemical stain for desmin, ×200).

**Figure 10 cancers-17-01327-f010:**
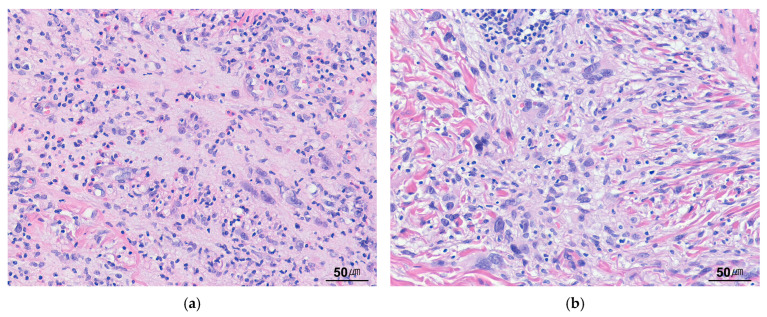
Myxoinflammatory fibroblastic sarcoma. (**a**) Atypical fibroblastic cells and a mixed inflammatory cell infiltrate are present in myxoid and hyaline stroma (H&E stain, ×200). (**b**) Tumor cells have macronucleoli (H&E stain, ×200).

**Figure 11 cancers-17-01327-f011:**
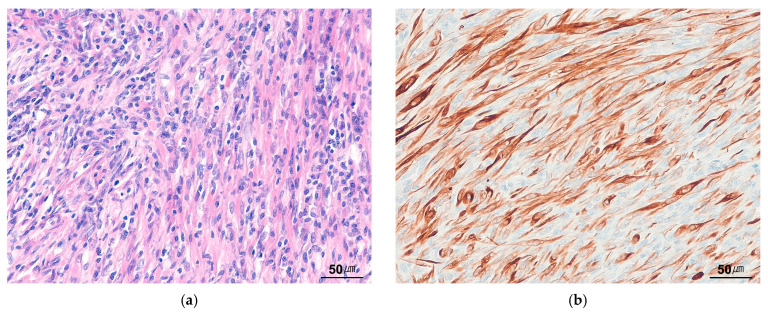
Inflammatory leiomyosarcoma. (**a**) Eosinophilic spindle cells with blunt-ended, elongated nuclei are arranged in a fascicular pattern, with a prominent lymphoplasmacytic infiltrate (H&E stain, ×200). (**b**) The tumor cells are positive for desmin (immunohistochemical stain for desmin, ×200).

**Table 1 cancers-17-01327-t001:** Histological patterns and atypical features of inflammatory myofibroblastic tumor.

Histological Pattern	Key Histological Features	Differential Diagnosis
Myxoid ^(a)^	Loosely arranged plump or spindled myofibroblasts; edematous myxoid background; abundant blood vessels; infiltrate of plasma cells, lymphocytes, and eosinophils	Inflammatory pseudotumor Inflammatory fibroid polyp Nodular fasciitis Myxoinflammatory fibroblastic sarcoma
Hypercellular	Compact spindle cells proliferation in fascicular or storiform patterns; variably myxoid and collagenous stroma; prominent inflammatory infiltrate.	Pseudosarcomatous myofibroblastic proliferation Inflammatory well-differentiated liposarcoma Low-grade myofibroblastic sarcoma Inflammatory leiomyosarcoma Embryonal rhabdomyosarcoma Gastrointestinal stromal tumor Epstein–Barr virus-positive inflammatory follicular dendritic cell sarcoma
Hypocellular fibrous	Low cellularity of spindle cells; hyalinized collagenous stroma; relatively sparse inflammatory infiltrate	IgG4-related sclerosing disease Desmoid fibromatosis
Atypical features ^(b)^	Hypercellularity, necrosis, ganglion-like cells, round or polygonal cells, herringbone pattern, multinucleated or anaplastic giant cells, atypical mitoses	

^(a)^ *ALK* rearrangements are not specific to histologic patterns of IMT and are observed in approximately 50–60% of IMTs. ^(b)^ Necrosis, hypercellularity, and ganglion-like cells are not associated with clinical outcomes. However, the presence of atypical mitoses raises concerns about the possibility of an alternative diagnosis.

**Table 2 cancers-17-01327-t002:** Molecular genetic alterations in inflammatory myofibroblastic tumor and epithelioid inflammatory myofibroblastic sarcoma.

	Genes Involved	Incidence	Cytogenetic Alterations	Gene Fusion	Immunohistochemistry
Inflammatory myofibroblastic tumor	*ALK* (2p23)	50–60%	t(1;2)(q22;p23)	*TPM3*::*ALK* ^(a)^	ALK (+) ^(d)^; SMA, calponin, h-caldesmon, desmin, cytokeratins (±)
			t(2;19)(p23;p13)	*TPM4*::*ALK*	
			t(2;17)(p23;q23)	*CLTC*::*ALK*	
			t(2;11)(p23;p15)	*CARS*::*ALK*	
			t(2;4)(p23;q21)	*SEC31A*::*ALK*	
			t(2;12)(p23;p11)	*PPFIBP1*::*ALK*	
			inv(2)(p23;q35)	*ATIC*::*ALK*	
			inv(2)(p21;p23)	*EML4*::*ALK*	
			t(2;17)(p23;q24)	*PRKAR1A*::*ALK*	
			inv(2)(p23;q34)	*FN1*::*ALK* ^(b)^	
	*ROS1* (6q22.1)	5–10%	t(3;6)(q12;q22)	*TFG*::*ROS1*	ROS1 (+)
			t(6;17)(q22;p13)	*YWHAE*::*ROS1*	
			t(2;6)(q35;q22)	*FN1*::*ROS1*	
			t(6;22)(q22;q12)	*TIMP3*::*ROS1*	
			t(6;7)(q22;p13)	*NUDCD3*::*ROS1*	
	*NTRK3* (15q25.3)	5–10%	t(12;15)(p13;q25)	*ETV6*::*NTRK3*	Pan-TRK (+)
	*RET* (10q11.21)	Rare	t(10;22)(q11;q12)	*TIMP3*::*RET* *FN1*::*RET* ^(c)^	
	*PDGFRB* (5q32)	Rare	t(5;12)(q32;q13)	*NAB2::PDGFRB*	PDGFRB (+)
Epithelioid inflammatory myofibroblastic sarcoma	*ALK* (2p23)	Rare	inv(2)(p23;q13)	*RANBP2*::*ALK* *RANBP1*::*ALK*	ALK (+) ^(e)^

^(a)^ Among inflammatory myofibroblastic tumors (IMTs) in the head and neck, *ALK* fusion with *TIMP3* has been most commonly detected. ^(b)^ Most urinary tract IMTs have recently been shown to harbor an *FN1*::*ALK* fusion. ^(c)^ Recently, an ALK-negative urinary bladder IMT with a novel *FN1*::*RET* gene fusion has been detected. ^(d)^ ALK immunostaining pattern varies depending on fusion partner. *CLTC*::*ALK* exhibits a granular cytoplasmic pattern, whereas most other *ALK* fusion variants demonstrate a diffuse cytoplasmic pattern. ^(e)^ Epithelioid inflammatory myofibroblatic sarcoma (EIMS) with *RANBP2*::*ALK* shows a nuclear membranous staining pattern, whereas EIMS with *RRBP1*::*ALK* demonstrates a perinuclear accentuated cytoplasmic pattern. +, positive staining; ±; variable staining.

**Table 3 cancers-17-01327-t003:** Subtypes of inflammatory myofibroblastic tumor and recently described myofibroblastic sarcoma.

	Clinical Features	Histological Features	IHC	Molecular Genetics	Prognosis ^(a)^
Pregnancy- associated inflammatory myofibroblastic tumor	Occur incidentally in pregnant women; usually in the setting of prenatal complications such as DM, hypertension, preeclampsia, or abnormal implantation	Three patterns (1) hypocellular and myxoid, (2) decidualized and cellular with a moderate myxoid matrix, (3) smooth muscle-like with minimal myxoid stroma	ALK (+), PR (+), CD10 (+), SMA (±), ER (±)	*ALK* fusion with *TIMP3* and *THBS1*; fusion involving *RET*, *ROS*, or *INSR* (a subset of ALK-negative tumors)	Benign
Epithelioid inflammatory myofibroblastic sarcoma	Median age of 39 years; male predominance; intraabdominal region, particularly mesentery and omentum	Sheets of epithelioid and round cells with vesicular nuclei; abundant myxoid stroma; predominant neutrophilic infiltration	ALK (+) ^(b)^, desmin (+), CD30 (+), SMA (±)	*RANBP2*::*ALK*; *RRBP1*::*ALK*	More aggressive
Myxoid inflammatory myofibroblastic sarcoma	Median age of 37 years; affects males and females equally; predominantly occurs in peritoneum, but also in paratesticular region, chest wall, extremity, retroperitoneum, uterus	Bland to mildly atypical neoplastic myofibroblasts; delicate branching vessels; myxoid stroma; infiltrative growth; inflammatory infiltrate	SMA(±), desmin (±), CD30 (±), CD34 (±); rare expression of ALK (4% of cases)	*PDGFRB* rearrangements*; PML*::*JAK1*; *SEC31A*::*PDGFRA*; *KRAS* mutations (G12V and Q61H) (a small subset)	More aggressive

^(a)^ Compared with the prognosis of conventional IMT. ^(b)^ *RANBP2*::*ALK* is associated with nuclear membranous staining pattern, whereas *RANBP1*::*ALK* shows perinuclear accentuated cytoplasmic pattern. IHC, immunohistochemistry; ALK, anaplastic lymphoma kinase; SMA, smooth muscle actin. +, positive staining; ±, variable staining.

**Table 4 cancers-17-01327-t004:** Differential diagnosis of inflammatory myofibroblastic tumor: clinical, histological, immunophenotypic, and molecular features.

Tumor Type	Clinical Features	Histological Features	IHC	Molecular Features
Inflammatory pseudotumor	Associated with infection, immune reaction, post-traumatic	Mixed inflammatory cells infiltrate; fibroblasts, myofibroblasts; fibrosis	SMA (+); ALK (−)	No known genetic alterations
IgG4-related sclerosing disease	Middle-aged to elderly adults; elevated serum IgG4 levels in most cases	Dense lymphoplasmacytic infiltrate; storiform fibrosis; obliterative phlebitis	IgG4-positive plasma cells	No known genetic alterations
Pseudosarcomatous myofibroblastic proliferations	Previous instrumentation or surgery; genitourinary tract, especially urinary bladder	Myofibroblasts with palely eosinophilic cytoplasm; tissue culture-like pattern	SMA (+); desmin (+); ALK (+) in 70%	*FN1*::*ALK*
Inflammatory fibroid polyp	Middle-aged adult; slight female predominance; stomach, small intestine	Spindled cells; edematous or myxoid stroma; inflammatory infiltrate, prominent eosinophils	CD34 (+); SMA (±); desmin (±)	*PDGFRA* activating mutations
Nodular fasciitis	Young adults; rapidly grow; subcutaneous tissue of the extremity	Myofibroblastic proliferation; tissue culture-like growth pattern; microcystic changes	SMA (+), often diffuse; desmin (±)	*USP* rearrangement
Desmoid fibromatosis	Young to middle-aged adults, pediatric; abdominal wall, extremities, mesentery	Uniform fibroblastic proliferation; long sweeping fascicles. collagen deposition	SMA (+); desmin (±); nuclear β-catenin (+) in majority	Somatic mutations in *CTNNB1*; germline mutations in *APC*
Inflammatory well-differentiated liposarcoma	Older adults; mostly arise in retroperitoneum	Dense chronic inflammatory cells; scattered atypical stromal cells; lipoblasts	MDM2 (+); CDK4 (+)	*MDM2* and *CDK4* amplification
Low-grade myofibroblastic sarcoma	Adults; head and neck, especially tongue, oral cavity, extremities	Cellular fascicles of spindle cells; diffusely infiltrative growth	SMA (±), desmin (±)	Alterations of 12p11 and 12q13-q22 in some cases
Myxoinflammatory fibroblastic sarcoma	Adults; acral dorsal extremities, particularly fingers, hands	Atypical fibroblastic cells with macronucleoi; myxohyaline stroma; inflammatory infiltrate	CD34 (±); SMA (±); CD68 (±)	t(1;10)(p22;q24); *TGFBR3*::*OGA; BRAF* rearrangements
Inflammatory leiomyosarcoma	Adults; inflammatory-type symptoms; retroperitoneum, lower limb, trunk, lung	Fascicles of atypical spindle cells; prominent, diffusely inflammatory infiltrate	SMA (+); desmin (+); myogen (±)	Near-haploid genotype
Embryonal rhabdomyosarcoma	Children and young adults; head and neck, genitourinary system	Primitive round and spindle cell morphology; scattered rhabdomyoblasts	Desmin (+), myogenin (+), MYOD1 (+)	Complex aneuploid karyotype
Gastrointestinal stromal tumor	Gastrointestinal tract; mesentery, omentum, retroperitoneum	Broad morphologic spectrum; spindle cell, epithelioid, or mixed morphology	CD117 (+), DOG1 (+); PDGFRA (+) ^(^^a)^; SDHB loss ^(b)^	*KIT* or *PDGFRA* mutations
EBV-positive inflammatory follicular dendritic cell sarcoma	Young to middle-aged adults; Occurs almost exclusively in liver, spleen	Atypical spindle to oval cells; accompanied by a rich lymphoplasmacytic infiltrate	CD21 (+), CD23 (+), CD35 (+)	Copy-neutral loss of heterozygosity of 5q; gain of chromosome X

IHC, immunohistochemistry; ALK, anaplastic lymphoma kinase; SMA, smooth muscle actin; MYOD1, Myogenic differentiation 1; PDGFRA, Platelet-derived growth factor receptor alpha; SDHB, succinate dehydrogenase subunit B; EBV, Epstein–Barr virus. +, positive staining; ±, variable staining. ^(a)^ PDGFRA-mutant gastrointestinal stromal tumors (GISTs) typically show strong and diffuse expression of PDGFRA. ^(b)^ Succinate dehydrogenase-deficient GISTs show loss of staining for SDHB.

## Data Availability

All data were included in the manuscript.

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
