# Peer review of "Inflammatory Myofibroblastic Tumor: An Updated Review"

_cancers, 2025, doi:10.3390/cancers17081327_

Round 1
Reviewer 1 Report
Comments and Suggestions for Authors
This review comprehensively analyzed the clinical, pathological, and molecular genetic characteristics of inflammatory miofibroblastic tumor, highlighting diagnostic approaches and crucial differential considerations.
Abstract disorganized. Would start describing IMT.
Would further developed the description of imaging findings and the differentiation with other entities. Also, a table with peculiar findings in each imaging technique would be an added value. Please remove tables 1 and 2.
Would switch paragraphs 4 and 5/6. Histology comes before genetiucs.
Table 4 should be detailed further.
Would summarize paragraphs on differential diagnosis. Too long.
Treatment should be developed further.
Comments on the Quality of English Language
Many grammar and syntax errors.
Author Response
This review comprehensively analyzed the clinical, pathological, and molecular genetic characteristics of inflammatory myofibroblastic tumor, highlighting diagnostic approaches and crucial differential considerations.
Abstract disorganized. Would start describing IMT.
-> Thank you for your valuable comment. I have deleted the sentence “Primary intrathoracic sarcomas constitute a rare, heterogeneous group of neoplasms that occur in the lung parenchyma, pleura, and mediastinum.”
Would further developed the description of imaging findings and the differentiation with other entities. Also, a table with peculiar findings in each imaging technique would be an added value.
-> Thank you for your suggestion. I have expanded the description of IMT’s imaging characteristics as follows.
“These lesions demonstrate heterogeneous enhancement on computed tomography (CT). … Retroperitoneal IMTs typically reveal an isointense to slightly hypointense signal on T1-weighted images and a hyperintense signal on T2-weighted images, depending on the cellular composition, and are usually accompanied by heterogeneous contrast enhancement. Differentiating IMT from inflammatory lesions and other soft tissue neoplasms based solely on imaging findings is challenging. Accurate diagnosis requires integration of imaging with clinical evaluation and laboratory test results.”
Please remove tables 1 and 2.
-> Thank you for your suggestion. Tables 1 and 2 have been removed.
Would switch paragraphs 4 and 5/6.
Histology comes before genetics.
-> Thank you for your comment. I have switched paragraphs 4 and 5/6.
Table 4 should be detailed further.
-> Thank you for your comment. The revised Table 1 (previously Table 4) is more detailed.
Would summarize paragraphs on differential diagnosis. Too long.
-> Thank you for your suggestion. I have shortened the differential diagnosis paragraph to improve clarity and conciseness.
Treatment should be developed further.
-> Thank you for your suggestion. I have developed the treatment paragraph.
Reviewer 2 Report
Comments and Suggestions for Authors
The author provides a comprehensive and extensive review of inflammatory myofibroblastic tumor covering diagnosis, presentation, various differentials, prognosis, and treatment. The review is very well-written and appropriately referenced. The tables and figures are well-prepared and add useful and informative information for the reader.
Comments:
- I would suggest deleting the 1st sentence of the abstract reference primary intrathoracic tumors and lead, instead, with the second sentence that introduces IMT. The (current) 1st sentence suggests that the review is about primary intrathoracic tumors, when it is not.
- Minor comment - In various tables, there are several words that are hyphenated when they could just be moved to the next line in total. They are distracting and take away from the overall readability.
- Formatting issue - Table 4: Consider placing the words pattern, fibrous and features on their own lines, instead of listing as pat-tern, fi-brous, fea-tures with the 2nd half of each word on a different line.
- Formatting issue - Table 5: Consider having hyphenated words on their own line (i.e. in-flammatory, sar-coma, intra-ab-dominal, oc-curs, etc)
- Several words in Table 6 as well.
Author Response
The author provides a comprehensive and extensive review of inflammatory myofibroblastic tumor covering diagnosis, presentation, various differentials, prognosis, and treatment. The review is very well-written and appropriately referenced. The tables and figures are well-prepared and add useful and informative information for the reader.
Comments:
- I would suggest deleting the 1st sentence of the abstract reference primary intrathoracic tumors and lead, instead, with the second sentence that introduces IMT. The (current) 1st sentence suggests that the review is about primary intrathoracic tumors, when it is not.
-> Thank you for your suggestion. I have deleted the sentence “Primary intrathoracic sarcomas constitute a rare, heterogeneous group of neoplasms that occur in the lung parenchyma, pleura, and mediastinum.”
- Minor comment - In various tables, there are several words that are hyphenated when they could just be moved to the next line in total. They are distracting and take away from the overall readability.
- Formatting issue - Table 4: Consider placing the words pattern, fibrous and features on their own lines, instead of listing as pat-tern, fi-brous, fea-tures with the 2nd half of each word on a different line.
-> Thank you for your suggestion. I have placed the words pattern
- Formatting issue - Table 5: Consider having hyphenated words on their own line (i.e. in-flammatory, sar-coma, intra-ab-dominal, oc-curs, etc)
-> Thank you for your suggestion. I have hyphenated words on their own line.
- Several words in Table 6 as well.
- > Thank you for your suggestion. I have hyphenated words on their own line.
Reviewer 3 Report
Comments and Suggestions for Authors
In his manuscript entitled „Inflammatory Myofibroblastic Tumor: An Updated Review”, Dr. Choi reviews the current knowledge on the rare neoplasm Inflammatory Myofibroblastic Tumor (IMT). The author describes the epidemiology as well as clinical and radiological features of IMT in the different localizations, molecular features frequently associated with ALK rearrangements and the histopathology and immunohistochemical features of IMTs. Furtheron, the author describes diagnostic approaches and comprehensively comments on the differential diagnosis of IMTs in comparison to several morphologic mimics. Finally, the author comments on treatment and prognosis and future perspectives of the IMT. The manuscript is well written, comprehensive and covers an interesting topic. The literature is adequate and covers the statements made in the manuscript. I have only minor comments and recommend publication after the author has responded to them.
Here are my remarks in detail:
1) The histopathological and the immunhistochemical figures are lacking length bars. These would be preferable to assess the dimensions of the given preparations.
2) In the chapter “epidemiology”: it would be interesting to state something about the prevalence of IMT, if valid information are available.
3) Chapter 7 seems to be a bit out of context. The author should considering to fuse the information given here with another chapter. Same with chapter 10 and 11 - they could be fused together for one chapter “Treatment and prognosis”
4) Figure 6b and 6c: it might be an effect of differences in the cellular compositions, but these images do not seem to have the same dimensions, although the legend says that they are both in 200x magnification. Please re-check, whether both images are in 200x magnification.
5) Table 5 and 6 should be moved to the respective chapters.
6) l. 582 - 591: should be omitted, as not applicable here.
Finally, I want to thank the author for sharing his comprised knowledge with the medical community. Best regards.
Author Response
In his manuscript entitled „Inflammatory Myofibroblastic Tumor: An Updated Review”, Dr. Choi reviews the current knowledge on the rare neoplasm Inflammatory Myofibroblastic Tumor (IMT). The author describes the epidemiology as well as clinical and radiological features of IMT in the different localizations, molecular features frequently associated with ALK rearrangements and the histopathology and immunohistochemical features of IMTs. Further on, the author describes diagnostic approaches and comprehensively comments on the differential diagnosis of IMTs in comparison to several morphologic mimics. Finally, the author comments on treatment and prognosis and future perspectives of the IMT. The manuscript is well written, comprehensive and covers an interesting topic. The literature is adequate and covers the statements made in the manuscript. I have only minor comments and recommend publication after the author has responded to them.
Here are my remarks in detail:
1) The histopathological and the immunhistochemical figures are lacking length bars. These would be preferable to assess the dimensions of the given preparations.
-> Thank you for your suggestion. I have added length bars to the histopathological and immunohistochemical figures.
2) In the chapter “epidemiology”: it would be interesting to state something about the prevalence of IMT, if valid information are available.
-> Thank you for your comment. I have described it as follows.
“IMT is a rare mesenchymal neoplasm, accounting for less than 1% of all soft tissue tumors. Its exact prevalence is difficult to determine due to its rarity, histological heterogeneity, and historical misclassification as a reactive or inflammatory process”.
3) Chapter 7 seems to be a bit out of context. The author should considering to fuse the information given here with another chapter. Same with chapter 10 and 11 - they could be fused together for one chapter “Treatment and prognosis”
-> Thank you for your suggestion. Chapter 7 has been retained as a separate section to maintain the clarity of its specific content. However, Chapters 10 and 11 have been combined into a single chapter entitled “Treatment and Prognosis” to improve coherence and flow.
4) Figure 6b and 6c: it might be an effect of differences in the cellular compositions, but these images do not seem to have the same dimensions, although the legend says that they are both in 200x magnification. Please re-check, whether both images are in 200x magnification.
-> Thank you for your suggestion. I have discovered that Figure 6b was incorrectly labeled and was in fact 100× magnification. Figure 6b has been corrected to 100x magnification.
Table 5 and 6 should be moved to the respective chapters.
-> Thank you for your suggestion. Tables 5 and 6 have been moved to the appropriate chapters.
6) l. 582 - 591: should be omitted, as not applicable here.
-> Thank you for your suggestion. l. 582 – 591 have been omitted.
Finally, I want to thank the author for sharing his comprised knowledge with the medical community. Best regards.
-> Thank you for your kind words.
Reviewer 4 Report
Comments and Suggestions for Authors
This review article comprehensively describes the latest information on inflammatory myofibroblastic tumor (IMT), a rare soft tissue tumor. This article is of high quality; however, there is only one point on which I would like to request additional information, and I would appreciate your consideration.
MRI is the primary imaging modality in soft tissue tumors, including IMT. Since several coherent reports on IMT MRI findings exist, we would like to request the authors to add a description of characteristic MRI findings and also to add MRI images as Figures.
Author Response
This review article comprehensively describes the latest information on inflammatory myofibroblastic tumor (IMT), a rare soft tissue tumor. This article is of high quality; however, there is only one point on which I would like to request additional information, and I would appreciate your consideration.
MRI is the primary imaging modality in soft tissue tumors, including IMT. Since several coherent reports on IMT MRI findings exist, we would like to request the authors to add a description of characteristic MRI findings and also to add MRI images as Figures.
-> Thank you for your invaluable suggestion. Unfortunately, I was unable to locate any MRI images of IMT in my files or at my institution. I sincerely apologize for not being able to provide representative images. However, I have included a description of the characteristic MRI findings of IMT based on the published literature, as follows.
“… Retroperitoneal IMTs typically reveal an isointense to slightly hypointense signal on T1-weighted images and a hyperintense signal on T2-weighted images, depending on the cellular composition, and are usually accompanied by heterogeneous contrast enhancement. Differentiating IMT from inflammatory lesions and other soft tissue neoplasms based solely on imaging findings is challenging. Accurate diagnosis requires integration of imaging with clinical evaluation and laboratory test results.”
Round 2
Reviewer 1 Report
Comments and Suggestions for Authors
The paper was re-organized and significantly ameliorated following my previous suggestions.